# Combination Effect of Microcystins and Arsenic Exposures on CKD: A Case-Control Study in China

**DOI:** 10.3390/toxins15020144

**Published:** 2023-02-10

**Authors:** Hong Gao, Na Zhu, Shuxiang Deng, Can Du, Yan Tang, Peng Tang, Shuaishuai Xu, Wenya Liu, Minxue Shen, Xinhua Xiao, Fei Yang

**Affiliations:** 1Nursing Department, The Second Affiliated Hospital, Hengyang Medical School, University of South China, Hengyang 421001, China; 2School of Nursing, University of South China, Hengyang 421001, China; 3Department of Epidemiology and Health Statistics, The Key Laboratory of Typical Environmental Pollution and Health Hazards of Hunan Province, School of Basic Medicine, School of Public Health, Hengyang Medical School, University of South China, Hengyang 421001, China; 4Hunan Provincial Key Laboratory of Clinical Epidemiology, Department of Social Medicine and Health Management, Xiangya School of Public Health, Central South University, Changsha 410000, China; 5Department of Metabolism and Endocrinology, The First Affiliated Hospital, Hengyang Medical School, University of South China, Hengyang 421001, China

**Keywords:** arsenic, microcystins, chronic kidney disease, synergistic interaction

## Abstract

Evidence has shown that exposure to environmental pollutants such as microcystins (MCs), arsenic (As), and cadmium (Cd) can lead to the occurrence and development of chronic kidney disease (CKD). There is a synergistic effect between MCs and Cd. However, the combined effect of MCs and As exposures on CKD remains unclear. In Hunan province, China, 135 controls and 135 CKD cases were enrolled in a case-control study. Serum MCs, plasma As and Cd concentrations were measured for all participants. We investigated the association between MCs/As and CKD risk using conditional logistic regression. The additive model explored the interaction effect, and the Bayesian kernel machine regression (BKMR) models investigated the combined effects of MCs, As, and Cd on CKD. The results showed that MCs and As were significantly associated with CKD risk. Participants in the highest MCs concentration had a 4,81-fold increased risk of CKD compared to those in the lowest quartile (95% confidence interval [CI]: 1,96 to 11,81). The highest quartile of As concentrations corresponded to an adjusted odds ratio of 3.40 (95% CI: 1.51, 7.65) relative to the lowest quartile. MCs/As and CKD risk exhibited significant dose–response correlations (all *p* for trend < 0.01). In addition, a positive interaction effect of MCs and As on CKD was also reported. The CKD risk due to interaction was 2.34 times (95% CI: 0.14, 4.54) relative to the CKD risk without interaction, and the attributable proportion of CKD due to interaction among individuals with both exposures was 56% (95% CI: 0.22, 0.91). In the BKMR, the combined effect of MCs, As, and Cd was positively associated with CKD. In conclusion, both MCs and As are independent risk factors for CKD, exerting a synergistic effect between them. Combined exposure to MCs, As, and Cd can increase the risk of CKD.

## 1. Introduction

The estimated prevalence of chronic kidney disease (CDK) globally is 13.4% [1], with low- and middle-income nations bearing 63% of the disease burden [2]. A nationwide survey conducted between 2009 and 2010 has shown that the prevalence of CKD in China was 10.8% [3]. The National Health and Nutrition Examination Survey (NHANES) between 2011 and 2012 indicated that the prevalence rate of CKD was 31.5% in those 65–79 years old and 65.0% in those over 80 years old [4]. Due to the high prevalence rate of CDK, it has become a major public health problem worldwide.

The prognosis of CKD Is poor, so it is crucial to prevent its occurrence [5]. Previous investigations suggested that CKD arises from many heterogeneous diseases that alter renal function. Among them, genetic and environmental factors were two significant factors [5,6]. Many studies have shown that environmental pollutants are closely related to the risk of renal dysfunction [7,8,9], including the common pollutants microcystins (MCs) and heavy metals (arsenic [As] and cadmium [Cd]) [10,11]. Our previous study has shown that microcystin-LR (MC-LR) is an independent risk factor for CKD and has a synergistic relationship with Cd [12]. However, the effects of combined exposure to MCs and As on kidney injury and even CKD are unknown.

Lake eutrophication has spread worldwide, including 70% of lakes in China [13]. It leads to the frequent appearance of cyanobacteria, which produce a lot of MCs (e.g., MC-LR) [14,15,16,17]. As levels in drinking water are also well above the limits set by the World Health Organization in many countries and regions [18,19,20]. MCs and As can enter the human body mainly through contact with drinking water, soil, food, and other ways [21,22,23,24,25]. They accumulate in the kidney and are excreted from the body through the kidney [26,27]. Some studies have shown that exposure to MCs and As can lead to structural and functional changes in the kidney, leading to kidney injury and even the development and progression of CKD [28,29,30].

Molecular epidemiological studies have shown that exposure to low levels of MCs is associated with significant disruption of amino acid metabolism and a significant risk of kidney damage [31]. MCs can alter the renal damage markers of serum creatinine and uric acid in fishermen [32]. A cross-sectional study found that MCs exposure was associated with renal impairment [33]. As was a non-negligible risk factor for CKD and was linked to an increased CKD risk [34,35]. A dose-dependent association between As concentration and kidney diseases was observed [36]. MCs and As have widely coexisted in the environment for a long time [21]. Animal studies have shown that As and cyanobacterial co-exposure can reduce the infection-fighting ability of rainbow trout and increase the toxic effects [37]. However, no studies have reported the combined effect of MCs and As on human renal function.

In this study, we conducted a case-control study to investigate the single and combined effects of MCs and As, as well as the combined effects of MCs, As, and Cd on human CKD risk in central China.

## 2. Results

### 2.1. Characteristics of Study Population

This study included 135 CKD cases and 135 healthy individuals as participants. The average age of all the subjects was 59 ± 14.29 years old. The average age of the case group was 60.19 ± 1.22 years and the control group was 57.81 ± 1.24 years. Our previous study has described more details regarding the population characteristics of the cases and controls [12]. Except for hypertension (*p* < 0.05), there were no significant differences between the control group and case groups in age, sex, drinking, smoking, BMI, ethnicity, level of education, occupation, marital status, physical exercise, hyperlipidemia, or diabetes (all *p* > 0.05).

### 2.2. The Relationship between MCs and the Risk of CKD

Appendix A shows that the median exposure level of serum MCs in the total population was 0.16 μg/L. The control group (0.14 μg/L) had lower levels of serum MCs than cases (0.16 μg/L) [12]. The distribution of serum MCs concentrations at different estimated glomerular filtration rate (eGFR) levels is presented in Figure 1 and Appendix A. Serum MCs concentrations were different with different eGFR groups. The MCs concentration in the lowest eGFR group (0.20 μg/L) was significantly higher than in the other groups (*p* < 0.001).

The associations between serum MCs and CKD risk are presented in Table 1. Compared with the lowest concentrations of serum MCs, the estimated adjusted odds ratios (AORs) for groups Q2, Q3, and Q4 were 3.70 (95% CI: 1.38, 9.93), 4.20 (95% CI: 1.62, 10.90), and 4.81 (95% CI: 1.96, 11.81), respectively. There was a dose–response relationship between MCs exposure and the risk of CKD (*p* for trend < 0.05).

### 2.3. Association between As and CKD Risk

In our study population, the median exposure level of plasma As was 1.12 μg/L, and the case group (1.15 μg/L) was higher than the control group (1.00 μg/L) (see Appendix A). Figure 2 and Appendix A show the distribution of plasma As concentrations at different eGFR levels. The plasma As level was the lowest when eGFR ≥ 90 mL /min/1.73 m^2^ (0.93 μg/L), and it was the highest when eGFR < 30 mL/min/1.73 m^2^ (2.36 μg/L).

Table 2 shows the associations between plasma As and CKD risk. The estimated AOR for CKD was more than triple for the highest As quartile (AOR = 3.40; 95% CI: 1.51, 7.65) compared with the lowest quartile. Compared with the lowest quartile of As, the AORs for groups Q2 and Q3 were 0.89 (95% CI: 0.37, 2.11) and 2.51 (95% CI: 1.06, 5.96). A monotonic increase in AORs according to As quartile was also observed (*p* for trend < 0.05).

### 2.4. Combined Effect of MCs and As on the Risk of CKD

Table 3 showed the interactive effect between As and MCs on CKD risk. Compared with the combined low MCs and the low As, the AOR for the combined high MCs and the high As was 4.14 (95% CI: 1.97, 8.73). Based on the additive model, the relative excess risk due to interaction (RERI) and the calculated values of attributable proportion (AP) was 2.34 (95% CI: 0.14, 4.54) and 0.56 (95% CI: 0.22, 0.91), respectively. This means that the CKD risk due to interaction was 2.34 times relative to the CKD risk without interaction, and the attributable proportion of CKD due to interaction among individuals with both exposures was 56%. The RERI and AP were positive and statistically significant, which showed a positive interaction between MCs and As exposures, which can synergistically increase the risk of CKD.

### 2.5. Combined Effects of MCs, As, and Cd on CKD

MCs and CD had a synergistic effect [12]. So, we further used the BKMR models to estimate the combined effects of MCs, As, and Cd on CKD. Appendix A shows that the conditional PIPs were higher than 0.8 in these three exposures, and the conditional PIP of MCs were even higher, at 0.9991. The overall effect of the combined exposures is displayed in Figure 3. Compared to the 50th percentile of the MCs, As, and Cd mixture, a significant positive combined effect of this mixture at other percentiles on CKD was observed (Figure 3).

### 2.6. Sensitivity Analyses

The sensitivity analyses are presented in Appendix A. We observed a similar relationship between MCs/As and CKD risk diagnosed according to eGFR < 60 mL/min/1.73 m^2^, which showed the highest ORs in the highest As (AOR = 2.35; 95%CI: 1.09, 5.07) and MCs (AOR = 6.00; 95%CI: 1.85, 19.48) exposure groups. Dose−response relationships existed (*p* for trend < 0.05). However, the association between MCs/As and CKD was not statistically significant when grouped by the presence or absence of albuminuria.

## 3. Discussion

The present study first examined the relationship between MCs combined with As and CKD risk in humans. Our case-control study showed MCs and As are both risk factors for CKD with a dose−response relationship. Interestingly, there was a positive association and interaction between exposure to MCs and As on CKD risk. Meanwhile, there was a significant positive combined effect of MCs, As, and Cd.

The contamination of MCs and As is becoming increasingly serious to the development of the social economy [38,39,40,41]. The median exposure level of MCs and As were 0.16 μg/L and 1.12 μg/L, respectively, indicating that subjects were exposed to MCs and As. There are several reasons for exposure to MCs and As. MCs pollution is widespread and serious in China, which can be detected in water, soil, and vegetables at the highest concentrations of 514 μg/L, 187 μg/kg, and 382 μg/kg, respectively [23]. Hunan province is located in the center of China, which has a subtropical monsoon climate with high temperatures and strong photosynthesis, and is suitable for MCs [42,43]. In recent years, local cyanobacteria blooms have occurred in Hunan Province, with a risk of large-scale blooms [44]. Our previous research found high levels of cyanobacteria in freshwater ponds and the surface water of Dong Ting Lake in Hunan Province, which can produce cyanotoxins such as MC-LR, MC-RR, and MC-YR [44,45]. Besides, Hunan areas have abundant mineral resources, including realgar, coal, and other minerals, leading to the severe excess of As content in the ecological environment and agricultural products [46,47]. Previous studies in this area have found that the concentration of arsenic in top soils, surface water, and food can reach 99.51 mg/kg, 10,400 μg/L, and 0.16mg/kg, respectively [24,48].

Our study confirmed that MCs and As exposure were independent risk factors for CKD and showed a dose−response relationship with CKD risk, consistent with other studies. At present, As is a recognized risk factor for CKD. A long-term prospective observational study found a dose-dependent association between well-water As concentration and kidney diseases [35]. Residing closer and higher As and polycyclic aromatic hydrocarbon exposure were associated with the renal petrochemical industry [34]. Studies on the association between MCs and CKD are limited, but many studies believe that MC-LR can cause renal function decline and kidney injury in the population. MC-LR was found to cause changes in serum creatinine and uric acid markers in 35 fishers [32]. Chronic exposure to MC-LR reduced eGFR, leading to renal impairment [33]. Our previous study suggested that the risk of CKD increases with the increase in MC-LR concentration [12]. This study also indicated that MCs and As were significant risk factors for the decline of eGFR levels. The total eGFR level decrease in CKD patients can reduce the ability to remove xenobiotics and toxicants, resulting in the continuous accumulation of toxic substances in the blood, thus increasing the burden on the kidney and forming a vicious circle [9].

MCs and As contamination co-exist in the water environment for a long time and must be monitored continuously. However, there are no epidemiological studies on the health hazards of MCs and As combined pollution. Interestingly, we are the first to find a synergistic effect between MCs and As on CKD. This co-exposure can synergistically increase the risk of CKD. Some research has found that As and cyanobacteria co-exposure can decrease the ability to control infection and cause changes in the plasmatic parameters to enhance the effects on rainbow trout [37,49]. A study on the health risks of farmed tilapia found that the presence of MCs and As in the fish posed a toxic risk for consumption [21].

Some animal studies have indicated that MCs or As can accumulate in the kidney and cause nephrotoxicity [28,50,51]. MCs and As exposure can lead to structural and functional changes in the kidney, such as increased Scr and BUN and decreased eGFR [52,53]. Our previous study showed that MCs and Cd share common mechanisms of nephrotoxicity [12], and these are equally present in the mechanisms of As-induced nephrotoxicity. Both MCs and As exposure also can cause the oxidation–antioxidant system imbalance, which increases ROS/MDA/SOD/CAT levels and increases the levels of GSH, leading to lipid peroxidation and apoptosis of renal tubular epithelial cells [54,55,56]. Our previous animal experiments showed that MC-LR could induce kidney injury through the PI3K/AKT signaling pathway [12]. Some studies indicate that As could cause oxidative stress, inflammation, and renal apoptosis by activating the PI3K/AKT signaling pathway [57,58]. In addition, MCs and As both can increase the expression of Bax and p53 and enhance the activity of caspase-3 to induce oxidative stress and apoptosis of renal cells [50,58,59]. Although current studies have shown that MCs and As have some common toxic effects and mechanisms on the kidney, further experiments still are needed to explore and verify the combined mechanism, which is the key to the prevention and treatment of renal insufficiency and deterioration of CKD.

We first reported the association between MCs combined with As and CKD risk, as well as the combined effect of MCs, As, and Cd. These findings can offer clues for the epidemiological study of multiple environmental exposure and kidney disease and provide references for the national policy of environmental health and environmental quality standards. Our study has several limitations. Firstly, we can not measure all the variants of MCs by using an ELISA kit due to the cross-reactivity of MC antibodies with several naturally-occurring metabolic MC-detoxification products [60]. Furthermore, the combined effects of MCs and As must be validated through experimental studies in cells or animals. Finally, appropriate multicenter and prospective large-sample studies must be conducted to assess the association between the combination of MCs and As exposures and CKD risk.

## 4. Conclusions

We provided evidence that MCs and As were risk factors for CKD, and there was a positive synergistic interaction between MCs and As on CKD risk. A combined effect of MCs, As, and Cd existed on the risk of CKD. The findings suggest that governments should consider policies and appropriate intervention measures to prevent human and animal exposure to MCs and As, particularly in areas with abundant cyanobacterial blooms and mineral resources.

## 5. Materials and Methods

### 5.1. Study Population

We conducted this case-control study previously reported, in which participants were over 18 years old and had lived locally for more than 5 years [1]. In China, the Hunan province is rich in non-ferrous metal resources [61]. All participants in this study signed informed consent. We excluded participants undergoing surgery or taking drugs affecting kidney function and who were medically diagnosed with diabetic nephropathy or hypertensive nephropathy. We used the Chronic Kidney Disease Epidemiology Collaboration [CKD-EPI] equation to calculate the eGFR value in Chinese adults. CKD was defined as an eGFR < 60 mL/min/1.73 m^2^ and albuminuria [1]. Based on the participants in the previous cross-sectional study [30], 135 cases were screened based on the diagnostic criteria of CKD as well as inclusion and exclusion criteria. 135 controls were matched by age (±3 years), sex, and study area [12]. Finally, a total of 270 subjects were enrolled in this study. The study was approved by the Ethical Committee of Xiangya Hospital, Central South University.

### 5.2. Measurement of MCs Exposure

Serum MCs concentrations were detected by ELISA kits [(#20-0068), Beacon Analytical Systems Inc., Saco, ME, USA]. The testing process of serum MCs concentrations strictly followed the kit instructions, and the detailed steps were described in the reported study [12]. The steps were as follows: (a) Adding 50 µL of MCY-HRP enzyme conjugate solution to each well. (b) Preparing 50 µL standard solution (0.1, 0.3, 0.8, 1.0, and 2.0 µg/L) and negative control. (c) Adding 50 µL of each blood sample into the specified well. (d) Whirling microplate rapidly to mix the solution thoroughly, then incubating for 30 min at 37 °C in the dark. (e) Washing 5 times and then removing the remaining solution in wells by turning and tapping the plate on the absorbent paper. (f) Add 100 µL of substrate solution to each well and shake lightly after incubating for 30 min at 37 °C in the dark. (g) Adding100 µL of stop solution to each well in proper order. (h) Reading the plate on a microtiter at 450 nm. (i) Plotting a 5-point calibration curve for standard MC-LR at concentrations of 0.1~2.0 µg/L and plotting the absorbance of MC-LR, which corresponds to its concentration by log-linear regression. (j) Repeating all samples in duplicate and calculating average values. The detection limit of microcystin was 0.1–2 μg/L. We conducted a recovery test to verify the accuracy by adding a standard microcystin substance to the sample.

### 5.3. Measurement of As and Cd Exposures

Plasma concentrations of As and Cd were measured by using quadrupole inductively coupled plasma-mass spectrometry (ICP-MS; Agilent 7700× ICP-MS; Agilent Technologies, Santa Clara, CA, USA). To ensure the accuracy of ICP-MS, certified reference agents (ClinChek human plasma controls for trace elements No.8883 and 8884, Recipe, Munich, Germany) and standard reference materials 1640a (Trace Elements in Natural Water, National Institute of Standards and Technology, Gaithersburg, MD, USA) were analyzed in every 20 blood samples. Furthermore, two methods were used in our study for quality control of plasma metal determinations. One was three replicate measurements, and the other was the spiked recovery of pooled plasma samples (random sample extraction of 20 samples). The detection limit of As and Cd was 0.0055 μg/L and 0.0006 μg/L, respectively. If there were data below the detection limit in the sample, we would estimate the metal concentrations as half of the detection limit. It was accepted that the intra-assay and inter-assay coefficients of variation of plasma metals were less than 10%.

### 5.4. Statistical Analysis

The sample size was calculated using the classical statistical software PASS 15 for a matched case-control study. The estimated case sample size was 96. The management and analysis of all research data were mainly performed by SPSS software version 26.0 and R 3.6.1. The mean ± standard deviation (SD) was used to describe the normal distribution of quantitative data, and the P50 (P25, P75) was used to describe the non-normal distribution data. Numbers (percentages) were used to describe qualitative and semi-quantitative data. The eGFR levels were divided into groups 1, 2, 3, and 4 according to the ranges of <30 mL/min/1.73 m^2^, 30~59 mL/min/1.73 m^2^, 60~89 mL/min/1.73 m^2^, and ≥90 mL/min/1.73 m^2^, respectively. The differences in some variables between the case and control groups were analyzed by *t*-test, Chi-square test, or nonparametric test. We divided MCs and As concentrations into groups Q1, Q2, Q3, and Q4 according to the quartiles of the control group. The associations between the levels of MCs, As, and the risk of CKD were analyzed by conditional logistic regression. After dividing participants into low and high concentrations according to the median values of MCs and As concentrations in the control group, we used an additive model to study the interaction effect and calculated the interaction-related indicators, including RERI and AP. RERI means the excess risk due to interaction relative to the risk without interaction and AP means the attributable proportion of CKD due to interaction among individuals with both exposures. If there was no interaction, the 95% CI of RERI /AP would include 0 or RERI/AP = 0. If both AP and RERI were >0, a positive synergistic interaction would exist. Our previous study showed that MCs have a synergistic effect with Cd [12]. So, we fitted Bayesian kernel machine regression (BKMR) models to further investigate the combined effects of MCs, As, and Cd on CKD after adjusting for hypertension. The posterior inclusion probability (PIP) for MCs, As, and Cd was estimated, with higher values (closer to 1) indicating greater importance. Furthermore, a sensitivity analysis was performed to explore the possible effects of different outcome definitions based on the presence of albuminuria and eGFR < 60 mL/min per 1.73 m^2^. All statistically significant analyses were two-sided and *p* < 0.05.

## Figures and Tables

**Figure 1 toxins-15-00144-f001:**
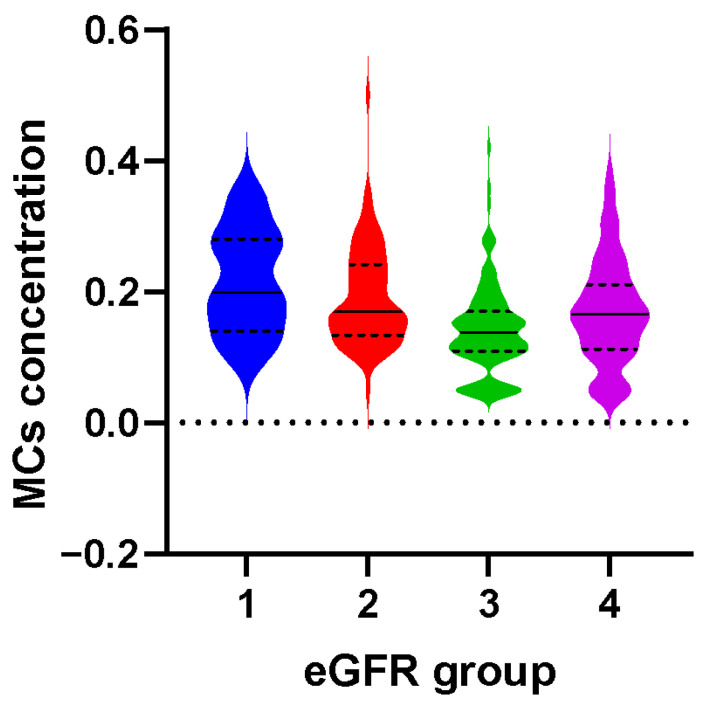
The distribution of serum MCs concentrations at different eGFR levels. The eGFR levels were divided into groups 1, 2, 3, and 4 according to the range <30 mL/min/1.73 m^2^, 30~59 mL/min/1.73 m^2^, 60~89 mL/min/1.73 m^2^, and ≥90 mL/min/1.73 m^2^, respectively. The number of participants in eGFR group 1, 2, 3, and 4 was 7, 78, 118, and 67, respectively.

**Figure 2 toxins-15-00144-f002:**
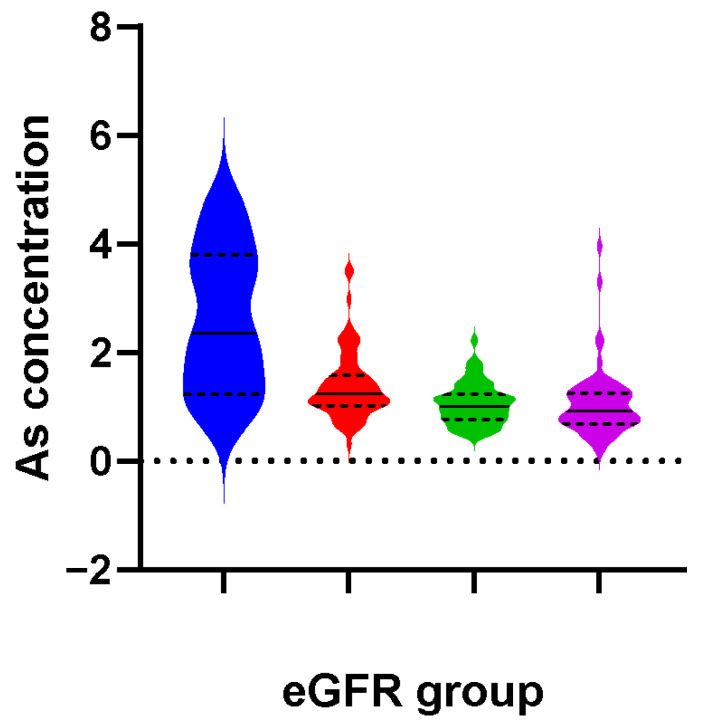
The distribution of plasma As concentrations with different eGFR levels. The eGFR levels were divided into groups 1, 2, 3, and 4 according to the range < 30 mL/min/1.73 m^2^, 30~59 mL/min/1.73 m^2^, 60~89 mL/min/1.73 m^2^, and ≥ 90 mL/min/1.73 m^2^, respectively. The number of participants in eGFR group 1, 2, 3, and 4 was 7, 78, 118, and 67, respectively.

**Figure 3 toxins-15-00144-f003:**
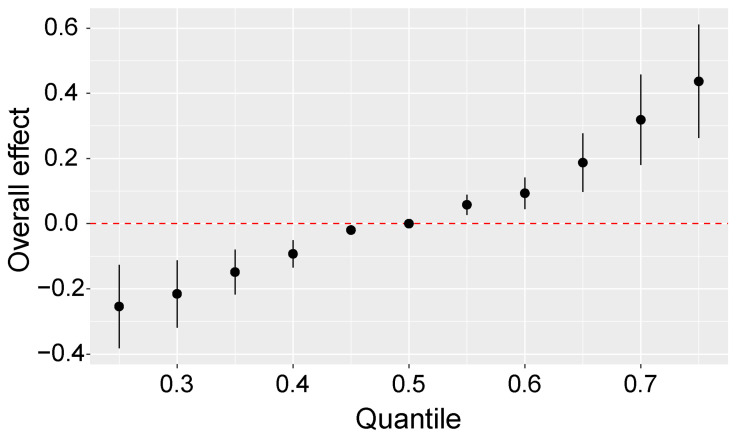
Overall effect of three exposures mixture estimates and 95% credible interval by Bayesian Kernel Machine Regression model, adjusted for hypertension. The exposures at the 50th percentile were set as the reference of the other percentiles.

**Table 1 toxins-15-00144-t001:** Association between MCs and risk of CKD using conditional logistic regression.

	Case/Control	OR (95% CI)	AOR (95% CI)
MCs			
Q1 (<0.10 μg/L)	10/33	1.0 (reference)	1.0 (reference)
Q2 (0.10 μg/L ~)	31/34	3.10 (1.27, 7.58)	3.70 (1.38, 9.93)
Q3 (0.14 μg/L ~)	44/34	4.14 (1.72, 9.95)	4.20 (1.62, 10.90)
Q4 (≥0.19 μg/L)	50/34	4.05 (1.79, 9.16)	4.81 (1.96, 11.81)
*p* for trend		0.003	0.003

The adjusted odds ratios (AORs): Adjusted for hypertension and As concentration. The concentrations of MCs were divided into groups Q1, Q2, Q3, and Q4 according to the quartiles of the control group, which were <0.10 μg/L, 0.10~ μg/L, 0.14~ μg/L, and ≥ 0.19 μg/L, respectively.

**Table 2 toxins-15-00144-t002:** Association analyses between As and risk of CKD using conditional logistic regression.

	Case/Control	OR (95% CI)	AOR (95% CI)
As			
Q1 (< 0.75 μg/L)	20/34	1.0 (reference)	1.0 (reference)
Q2 (0.75 μg/L ~)	23/34	1.06 (0.49, 2.30)	0.89 (0.37, 2.11)
Q3 (1.00 μg/L ~)	41/34	2.31 (1.07, 5.02)	2.51 (1.06, 5.96)
Q4 (≥ 1.31 μg/L)	51/33	2.94 (1.39, 6.24)	3.40 (1.51, 7.65)
*p* for trend		0.003	0.001

The adjusted odds ratios (AORs): Adjusted for hypertension and MCs concentration. The concentrations of As were divided into groups Q1, Q2, Q3, and Q4 according to the quartiles of the control group, which were <0.75 μg/L, 0.75~ μg/L, 1.00~ μg/L, and ≥ 1.31 μg/L, respectively.

**Table 3 toxins-15-00144-t003:** The combined effects of As with MCs on the risk of CKD in the additive model.

Variables	Case/Control	AOR (95% CI)	*p*	RERI (95% CI)	AP (95% CI)
MCs & As				2.34 (0.14, 4.54)	0.56 (0.22, 0.91)
Low & Low	16/31	Reference			
Low & High	25/36	1.40 (0.63, 3.11)	0.41		
High & Low	27/37	1.40 (0.64, 3.09)	0.40		
High & High	67/31	4.14 (1.97, 8.73)	<0.001		

The adjusted odds ratios (AORs): Adjusted for hypertension. “Low & Low” refers to participants with low MCs (<0.14 μg/L) and low As (<1.00 μg/L) concentration; “Low & High” refers to participants with low MCs (<0.14 μg/L) and high As (≥1.00 μg/L) concentration; “High & Low” refers to participants with high MCs (≥0.14 μg/L) and low As (<1.00 μg/L) concentration; “High & High” refers to participants with high MCs (≥0.14 μg/L) and low As (≥1.00 μg/L) concentration.

## Data Availability

The data that support the findings of this study are available from the corresponding author upon reasonable request.

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
