# Peer review of "Combination Effect of Microcystins and Arsenic Exposures on CKD: A Case-Control Study in China"

_toxins, 2023, doi:10.3390/toxins15020144_

Round 1

Reviewer 1 Report

This paper presents original datasets on effects of combination of MC and As on chronic kidney disease in China. The topic of this article is interesting. However, the manuscript needs some further improved before to be accepted for publication. Please check some specific comments that might be helpful of the authors to enhance the quality of the manuscript with grammatic issues.

Line27: The estimated global prevalence of chronic kidney disease (CDK) à The global prevalence of chronic kidney disease (CDK) globally

Line32: Due to its high prevalence à Due to high prevalence rate of CDK

Line59: Central China à central China

In Results section 2.1., please describe information between groups (case group vs. control group) in authors’ study.

Line61: Study Population Characteristics à Characteristics of Study Population

In the discussion section, in the detail, please describe prevalence rate of CDK and the severity of HABs in China.

Line130-132: Authors should find any previous study about cyanobacterial bloom and cyanotoxins in Hunan province.

Line 252 m2 à m2

Reviewer 2 Report

Journal: Toxins (ISSN 2072-6651)

Manuscript ID: toxins-2082659

Type: Article

Title: Effects of co-exposure to microcystin-LR and arsenic on chronic kidney disease: A Case-Control Study in Central China

Section: Marine and Freshwater Toxins

Special Issue: Recent Advances in Microcystins

This manuscript conducted a population-based case-control study involving 135 chronic kidney disease (CKD) cases and 135 matched controls in central China and analyzed the effects of microcystins (MCs) alone as well as combined with arsenic (As). In general, this topic is interesting. However, there are still some issues need to be addressed. I have the following comments and suggestions for the authors to improve the quality of manuscript.

1. Section “1. Introduction”

This is a following study of Feng et al. (2022) which analyzed the effects of MCs alone as well as combined with the known risk factor cadmium (Cd) on human CKd. Please introduce and cite the paper in the Introduction section.

Feng et al. 2022. Microcystin-LR Combined with Cadmium Exposures and the Risk of Chronic Kidney Disease: A Case-Control Study in Central China. Environmental Science & Technology, 56, 15818-15827. https://doi.org/10.1021/acs.est.2c02287

2. A published paper by He et al. (2022) also studied the effects of MCs on human kidney health. But the authors ignored this paper.

He et al., 2022. Health Risks of Chronic Exposure to Small Doses of Microcystins: An Integrative Metabolomic and Biochemical Study of Human Serum. Environmental Science & Technology, 6548-6559. https://doi.org/10.1021/acs.est.2c00973

3. Section “5. Materials and Methods”

Lines 206-209

“5.2. Measurement of MC-LR exposure

Serum MC-LR concentrations were detected by ELISA kits [(#20-0068), Beacon Analytical Systems Inc., USA]. The testing process of serum MC-LR concentrations strictly followed the kit instructions”

More than 279 MC derivatives have been reported. Method by enzyme-linked immunosorbent assay (ELISA) cannot determine the derivatives of MCs. Please read and cite the following paper. Please change the related expressions “MC-LR” to “MCs”. Please add some discussion for the limitations.

Please see and cite the following paper.

Chen et al., 2022. Challenges of using blooms of Microcystis spp. in animal feeds: A comprehensive review of nutritional, toxicological and microbial health evaluation. https://doi.org/10.1016/j.scitotenv.2020.142319

4. Section “2. Results”

Table S1

Please insert data of distribution of plasma MC concentration in the revised manuscript.

5. Lines 73-74

“The concentration of MC-LR in the lowest eGFR group was significantly higher than in the other groups (p<0.05).”

Please present values of MC concentrations in the revised manuscript.

6. Lines 90-91

“The plasma As level was the lowest when eGFR 90 mL /min/1.73m2, and it was the highest when eGFR < 30 mL/min/1.73m2.”

Please present values of As concentrations in the revised manuscript.

7. Lines 96-97

“A monotonic increase in AORs according to As quartile was also observed for these two compounds (p for trend < 0.05).”

What are these two compounds?

8. Lines 107-109

“Based on the additive model, the relative excess risk due to interaction (RERI) and the calculated values of attributable proportion (AP) was 2.34 (95% CI: 0.14, 4.54) and 0.56 (95% CI: 0.22, 0.91), respectively.”

What do RERI of 2.34 (95% CI: 0.14, 4.54) and the AP of 0.56 (95% CI: 0.22, 0.91 mean? Are there any threshold values for significance for RERI and AP? Please add some explanations in the revised manuscript.

9. Tables 1 and S2

For MCs, why values of OR for case/control and AOR of total population are different? Please add some explanations in the revised manuscript.

10. Tables 1 and S2

For As, why values of OR for case/control and AOR of total population are different? Please add some explanations in the revised manuscript.

11. Table S2

For MCs, why values of AOR of total population, group by eGFR, and group by albuminuria are different the values in Table S7 in Feng et al. (2022)? Both two studies use the same data. Please add some explanations in the revised manuscript.

12. Table S2

For As, why values of AOR of total population, group by eGFR, and group by albuminuria are different the values in Table S7 in Feng et al. (2022)? Both two studies use the same data. Please add some explanations in the revised manuscript.

13. What do the analyses of group by eGFR, and group by albuminuria mean? How to understand the analyses and data? Please add some description of results and explanations in the supplementary materials.

14. Reference list

The reference style does not follow the requirements by the journal Toxins. Please list all the authors of the cited papers.

15. Reference 24

It is Lin, H et al., 2016, but not Hui, L et al., 2016.

16. There are many grammatical and spelling errors and instances of badly worded/constructed sentences in the manuscript.

Reviewer 3 Report

Epidemiological data concerning exposure to cyanotoxins in humans are rare, which underlines the importance of this work for the scientific community. however, before publication, the manuscript requires small clarifications in order to improve its quality. It is important to underline the source of contamination of the populations in this study for arsenic and MC-LR. Is it mainly drinking water and if so what are the concentrations of these two contaminants in the study area.

minor comments:

- line 16: addictive effect and line 19: synergistic effect. In conclusion section it is indicated (line 186) a positive additive interaction between As and MC-LR. this needs to be checked and clarified if it is an additive or synergistic effect.

Round 2

Reviewer 2 Report

Journal: Toxins (ISSN 2072-6651)

Manuscript ID: toxins-2082659-peer-review-v2

Type: Article

Title: Combination Effect of microcystins and arsenic exposures on CKD: A Case-Control Study in China

Section: Marine and Freshwater Toxins

Special Issue: Recent Advances in Microcystins

This manuscript conducted a population-based case-control study involving 135 chronic kidney disease (CKD) cases and 135 matched controls in central China and analyzed the effects of microcystins (MCs) alone as well as combined with arsenic (As). However, there are still some issues need to be addressed. I have the following comments and suggestions for the authors to improve the quality of manuscript.

1. Section “5. Materials and Methods”

5.1. Study population

Lines 222-225

“According to the diagnostic criteria of CKD, 135 patients were screened, and controls were frequency-matched by age (± 3 years), sex, and study area. Finally, a total of 270 subjects, including 135 cases and 135 controls, were enrolled in this study.”

This is a following study of reference 30, Yang et al. (2019), which analyzed association of 23 metals in plasma and urine with kidney function. In the reference 30, a cross-sectional study was conducted to examine the relationship between exposure to multiple metals and renal-function indicators in residents in rural areas of Hunan province with rich mineral resources. A total of 3952 individuals were recruited. Of them, 3553 individuals completed both questionnaire interviews and clinical examinations. However, in this study, the number is 270. Why only 270 individuals were studied in this manuscript? Please describe the differences and reasons in the revised manuscript.

Ref. 30, Yang, F.; Yi, X.; Guo, J.; Xu, S.; Xiao, Y.; Huang, X.; Duan, Y.; Luo, D.; Xiao, S.; Huang, Z.; Yuan, H.; He, M.; Shen, M.; Chen, X. Association of plasma and urine metals levels with kidney function: A population-based cross-sectional study in China. Chemosphere, 2019. 226: 321-328.

Also, this is a following study of reference 12, Feng et al. (2022), which analyzed the effects of MCs alone as well as combined with the known risk factor cadmium (Cd) on human CKD. Please add analysis of combined effects of MCs, As and Cd.

Ref. 12, Feng, S.; Deng, S.; Tang, Y.; Liu, Y.; Yang, Y.; Xu, S.; Tang, P.; Lu, Y.; Duan, Y.; Wei, J.; Liang, G.; Pu, Y.; Chen, X.; Shen, M.; Yang, F. Microcystin-LR Combined with Cadmium Exposures and the Risk of Chronic Kidney Disease: A Case-Control Study in Central China. Environ Sci Technol, 2022. 56: 15818-15827.

2. Why analysis of sample size estimation was performed in the reference 30, Yang et al. (2019), in which, the estimated sample size is 3275, but analysis of sample size estimation was not performed in this study? Please add the differences and reasons in the revised manuscript.

3. Table S1

Please insert a column to show number of participants, including total, case and control.

4. Figures 1 and 2, Table S2

Please present number of participants in each eGFR group in the revised figures or tables.

5. Table 1

Why did you delete concentrations of MCs during the review process? Please insert the concentrations. How did you make the groups of Q1-Q4? Why number of cases are different, with 10 cases in Q1, 31 in Q2, 44 in Q3 and 50 in Q4? Please add some descriptions in the revised manuscript.

6. Tables 1 and 2

Why there is only a p trend? There are no p values for Q2, Q3 and Q4.

7. Lines 98-100

“The plasma As level was the lowest when eGFR 90 mL /min/1.73m2 (0.93 μg/L), and it was the highest when eGFR < 30 mL/min/1.73m2 (2.36 μg/L).”

Please cite Table S2 in this sentence.

8. Table 2

Why did you delete concentrations of As during the review process? Please insert the concentrations. How did you make the groups of Q1-Q4? Why number of cases are different, with 20 cases in Q1, 23 in Q2, 41 in Q3 and 51 in Q4? Please add some descriptions in the revised manuscript.

9. Table 3

How did you make the groups of Q1-Q4? Why number of cases and control are different, with 16 cases and 31 control in Q1, 25 and 36 control in Q2, 27 and 34 control in Q3, and 67 and 31 control in Q4? Please add some descriptions in the revised manuscript.

10. Table 3

What do you mean by low, high MCs and As? Please define it in the revised table.

11. Table 3

Why there are only p values for Q2, Q3 and Q4? There is no p trend.

12. Table 3

Lines 121-123

“The RERI and AP were positive and statistically significant, which showed a positive interaction between MCs and As exposures, which can synergistically increased the risk of CKD.”

No p values for RERI and AP are shown in the main text or table 3. How did you determine statistically significant?

13. Table S3

How did you make the groups of Q1-Q4? Please add some descriptions in the revised manuscript. Please present number of participants in each group, for total population, group by eGFR and albuminuria in the revised table.

14. Table S3

Why there is only a p trend? There are no p values for Q2, Q3 and Q4.

15. Reference list

Reference 32

It is Chen, J et al., 2009, but not Jun, C et al., 2009. First name and last names of all the authors are wrong. Also, references 32 and 50 are the same, while names of reference 50 are right.

16. Reference 33

It is Lin, H et al., 2016, but not Hui, L et al., 2016. First name and last names of all the authors are wrong. I have made this comment last time, but the authors said they are right. Please carefully check it.

Author Response

Happy New Year! Please see the attachment.
